# Production of Clubroot Standards Using a Recombinant Surrogate to Overcome Natural Genetic Variability

**DOI:** 10.3390/plants12081690

**Published:** 2023-04-18

**Authors:** Anjana Patel, Roy Kennedy

**Affiliations:** Warwickshire College University Centre (WCUC), Agri-Tech Research Unit, Pershore College, Wick WR10 3JP, UK; apatel@warwickshire.ac.uk

**Keywords:** clubroot standard, qPCR, recombinant clubroot DNA

## Abstract

Clubroot is caused by the obligate pathogen *Plasmodiophora brassicae*. The organism targets root hair cells for entry and forms spores in numbers so large that they eventually develop characteristic galls or clubs on the roots. Clubroot incidence is rising globally and impacting the production of oil seed rape (OSR) and other economically important brassica crops where fields are infected. *P. brassicae* has a wide genetic diversity, and different isolates can vary in virulence levels depending on the host plant. Breeding for clubroot resistance is a key strategy for managing this disease, but identifying and selecting plants with desirable resistance traits are difficult due to the symptom recognition and variability in the gall tissues used to produce clubroot standards. This has made the accurate diagnostic testing of clubroot challenging. An alternative method of producing clubroot standards is through the recombinant synthesis of conserved genomic clubroot regions. This work demonstrates the expression of clubroot DNA standards in a new expression system and compares the clubroot standards produced in a recombinant expression vector to the standards generated from clubroot-infected root gall samples. The positive detection of recombinantly produced clubroot DNA standards in a commercially validated assay indicates that recombinant clubroot standards are capable of being amplified in the same way as conventionally generated clubroot standards. They can also be used as an alternative to standards generated from clubroot, where access to root material is unavailable or would take great effort and time to produce.

## 1. Introduction

Clubroot disease caused by the obligate pathogen *Plasmodiophora brassicae* Woronin is a serious threat to the *Brassica* family [1]. Infection by this soil-borne pathogen leads to stunted growth and low yields. The spores of *P. brassicae* have a reported half-life of 3.4 years but can remain viable in soils for up to 15 years [2], making clubroot disease difficult to eradicate.

Dormant spores germinate as a response to growing *Brassica* roots to form primary zoospores. These primary zoospores breach the root wall and invade the root cells, diverting water and nutrients to further spore production and causing roots to swell into galls. This diversion of water and nutrients causes the host plant to display the typical symptoms associated with clubroot such as stunted growth and discolouration of leaves.

*P. brassicae* can also invade the roots of non-*Brassicae* hosts without causing further secondary infection [3].

The clubroot pathogen is difficult to eradicate due to the survival of spores in soil, with infections tending to cluster in patches [4]. The transmission of clubroot resting spores between sites arises via contaminated machinery entering previously uncontaminated sites [5]. The consumption of infected plants by livestock and wind-blown spores have also previously contributed to the transmission of the disease [6]. Previous studies using controlled glasshouse and field experiments have demonstrated that the disease development and severity of clubroot are both increased when temperature is increased. Climate warming may thus be a growing factor in increased clubroot incidence and severity [7].

Traditional clubroot control methods use crop rotations and lime treatment [8,9,10]. Breeding resistant cultivars however is the primary clubroot management technique, although several recent strains are now exhibiting resistance [11]. The continued use of clubroot-resistant crops, such as canola, are now being discouraged in fields with severe clubroot infections [12]. This is particularly problematic for the global production of oil seed rape (OSR), also a member of the *Brassicae* family. Oil seed rape (OSR) is a high-economic-value crop, and improving clubroot detection can help to reduce long-term crop loss.

Clubroot is one of the top pathogens affecting the production of OSR in Australia, Europe, China and Canada [13]. Yield losses depend on the severity of infection, with a higher degree of disease being positively correlated with a higher yield loss [14]. Reported field losses have exceeded 50% in Scotland, UK [15]. Estimations of 60% yield losses have been found for soil inoculated in glasshouses with a concentration range of 10^6^–10^8^ [16].

Early approaches to detecting clubroot were based on the appearance of “clubbed roots” as the central diagnostic feature, which defined clubroot presence [17]. This observational methodology was used for a long period of time due to the presence of organic substances, such as humic and fulvic acids, in soils. Whilst an essential component of soil structure, the acids inhibited the activity of polymerases in detection assays and prevented the accurate detection of clubroot. Newer approaches to overcome soil inhibitors use competitive internal positive controls to quantify the amount of inhibition, greatly improving the accuracy of the produced results [18]. Overcoming the inhibitory barriers to detection accelerated molecular biology techniques, particularly quantitative Polymerase Chain Reaction (qPCR). Alternative methods to detect clubroot involve the staining of infected root hairs to estimate disease severity [3]. Positive serological detection has also been reported [19].

Clubroot detection using PCR reduced the time taken to diagnose the disease, taking 24 h compared to the 1 to 2 months previously taken in bait testing. Clubroot genome regions that have been successfully amplified using this technique include the internal transcribed spacer region (ITS) [20], 18S ribosomal RNA [21] and the isopentyltransferase gene [22]. PCR accelerated the identification capability of plant disease diagnostics due to primer specificity but lacked a quantitative element, which is important for understanding disease incidence and following treatment programs.

qPCR eventually superseded PCR as the primary technique for quantifying clubroot. This method precisely analyses the number of clubroot spores present in a field from unknown samples and is currently considered the “gold standard” in clubroot testing. Improvements to the qPCR assay are constantly advancing the sensitivity and accuracy of produced data. A newer form of qPCR known as digital drop PCR (ddPCR) has demonstrated a high level of sensitivity when compared to qPCR [23].

The speed and sensitivity of qPCR assays enabled clubroot incidence to be managed more effectively. In qPCR assays, a standard curve is produced using a set of known clubroot concentrations. This is quantitatively analysed to determine the level of clubroot spores in unknown infected soil samples. Commercial diagnostic companies use qPCR to quantify clubroot levels in soil samples. Clubroot galls at different stages in their maturity have been regularly used to make standards; however, these have differences in their resting spore composition, as well as in their maturity. Variations in the use of clubroot gall tissues to make standards have potentially caused errors in the quantification of clubroot resting spores in samples. 

Several models have been developed to predict disease severity using resting clubroot spore levels in soil [24,25,26]. 

However, these studies lack the consideration of spore viability due to differences in the clubroot lifecycle and biotypes. The diverse nature of the clubroot genome varies across regions and countries, so a local strain may itself be composed of a mix of strains [27]. Some reports suggest that clubroot DNA is susceptible to degradation over time [28]; however, longer-term studies dispute this and provide data supporting the stability of clubroot DNA over time [29]. This work details a methodology that can be employed to overcome this variation.

Other issues that affect the quality of clubroot standard preparation include differences in the harvesting time of gall samples. Inconsistent harvesting times may lead to differences in the number of spores released into soil. Galls are made up entirely of viable and non-viable spores. These are then carried forward into assays and compared against unknown samples containing both viable and non-viable spores. Propidium monoazide, a DNA binding dye, has been shown to improve estimated clubroot spore viability [30]. Soil extraction kits may also contribute to DNA losses, which may then result in a lower amount of DNA being carried forward into actual assays. This is an issue particularly for the lowest concentration points because the spore number has a lower starting amount.

There is no existing standardised approach that is used to quantify clubroot standards because there is no test that can be used to standardise clubroot gall tissues. 

This has hampered molecular epidemiological and genetic studies on clubroot for 30 years. 

As a result, little or no useful comparable information on clubroot development and control has been possible to obtain. It has caused confusion, often with different results derived from similar studies. 

These differences in standard generation reduce the accuracy of diagnostic tests and risk misdiagnosing disease severity. Using conserved sequence clubroot standards will help to mitigate these differences, as only viable DNA is included. The production of recombinant clubroot conserved regions can help move towards a “standard” that can be used across various genomes to improve the accuracy of disease detection. Such a standard would enable the reliable quantification of clubroot spores in soil samples, aiding in the management and control of clubroot infections.

*Again, as a reminder, this work basically compares clubroot DNA standards produced in a recombinant vector against clubroot DNA standards produced from natural gall samples*. 

## 2. Results

The sequence of the internal transcribed spacer region (ITS) was used to produce both the recombinant DNA sequence and primer sequences, these sequences had previously demonstrated successful amplification by other groups [31].

In initial experiments, the forward (ITS3) and reverse primers (ITS4) were tested on a series of clubroot contaminated soil samples to validate the primers at increasing concentrations. Figure 1 illustrates the amplicons generated from a series of clubroot spore concentrations ranging from 10^3^ to 10^7^ in a PCR reaction.

A pure band corresponding to the size of the ITS region can be seen across all lanes in the gel image. This demonstrates the specificity of the primers and their ability to bind to the ITS regions at increasing concentrations of clubroot DNA.

Following the successful validation of the primer set against increasing concentrations of clubroot DNA, the ITS3 and ITS4 primer sequences were tested in a PCR reaction against the recombinantly produced ITS region DNA. The DNA sequence of this region can be found in the Methods Section.

The recombinant ITS DNA was amplified in a PCR reaction with the ITS3 and ITS4 primers to check the amplicon produced was the correct size. Amplification of a single band at a size of 100bp, demonstrated by the single band in lane 3 of the gel image in Figure 2, highlights the purity of the recombinant ITS DNA sequence. The PCR product also verifies the efficacy of the ITS3 and ITS4 primers to bind to the recombinantly produced DNA sequence.

Recombinant ITS DNA is represented by the middle band (lane 2); this band contains ITS recombinant DNA only. Lane 3 contains the successfully amplified recombinant ITS region. Both ITS regions have a molecular size of 100 bp.

Figure 2 demonstrates the successful PCR amplification of the recombinant ITS region DNA, and it illustrates a single pure band at a size of 100 bp. The single amplicon produced is indicative of a high primer specificity for the ITS region. A second PCR was subsequently carried out to further test the specificity over a range of recombinant DNA concentrations in a commercial clubroot assay. 

The concentrations tested ranged from 0.4 mg/mL corresponding to the highest concentration down to 0.000004 mg/mL corresponding to the lowest concentration. As the recombinant DNA concentration increased, the equivalent spore number also increased. This increase in detected spore number again highlights the ITS3 and ITS4 primer specificity for the recombinant ITS region. A comparison of the spore numbers generated from the recombinant ITS DNA is illustrated by the graph in Figure 3. The absolute values are listed in Table 1.

Figure 2 and Figure 3 demonstrate the specificity of the recombinant ITS DNA.

The spore number increased as the concentration of the recombinant DNA increased, indicating that the recombinant DNA is sensitive to different clubroot spore concentrations and can be amplified at a varying range of concentrations. 

The values obtained using the clubroot standards compared to the values generated using the recombinant standard are illustrated by a graph in Figure 3. The recombinant ITS DNA is illustrated by the blue line in Figure 3. The equivalent commercial standard concentration is also included as a benchmark of how effective the recombinant DNA is at being detected and acts as an independent measure on the efficiency of the reaction, as evidenced by the orange line in Figure 3. 

Figure 3 below illustrates a comparison of the detection of clubroot DNA between the recombinant ITS DNA and the clubroot DNA analysed in a qPCR assay.

The different DNA standards had the same R^2^ value.

Two points of interest are the spore concentrations at 1569 and 11,171, which are lower than the commercial clubroot standard spore number. 

These data highlight that the sensitivity of the recombinant DNA is higher at the 1569 and 11171 data values when compared to the commercial standards, which are derived from natural clubroot sources.

## 3. Discussion

The clubroot pathogen is a significant problem for the farmers and growers of *Brassicaceae* crops worldwide. High levels cause significant yield losses and affect crop quality. Breeding for resistance to clubroot is a key strategy for managing this disease, but detecting resistance can be challenging. One of the main issues around clubroot resistance detection is the variability in the pathogen population. *P. brassicae* has a high genetic diversity, and different biotypes can have varying virulence levels and different resistance genes. This means that resistance to one strain may not confer resistance to others, and the efficacy of resistance can vary depending on the pathogen population in a particular area.

Resistance to clubroot involves multiple genes, and the inheritance of resistance can be influenced by various factors, such as the environment and plant age. This complexity makes it difficult to identify and select plants with desirable resistance traits. Additionally, not all plants with clubroot show symptoms.

To address these issues, there is a need for standardised methods for clubroot detection.

Clubroot standards are currently produced by extracting spores from infected gall tissues. There are considerable differences in the gall tissues used and in the harvesting time of resting spores. These variations have an impact on the clubroot standards produced, reducing the accuracy of diagnostic tests. Therefore, measuring clubroot resting spore concentrations in relation to resting spore production in plants may give an inaccurate representation of the clubroot resting spore numbers in soils. Additionally, resting spore populations formed in clubroot galls may contain variations in clubroot biotypes. This means that a true representation of the clubroot population in soil might be difficult to obtain and that an obtained representation might be inaccurate. 

Extensive work has been previously undertaken in correlating recombinantly produced standards to different levels of clubroot spore concentrations from galls and establishing predictive models. However, these results were not strongly correlated, and the difference could be explained by differences between lifecycle stages, spore viability and genetic diversity. 

These differences can also have an impact on the validity of the clubroot standards currently used in qPCR assays.

Plant-derived standards represent local populations of clubroot, and it is unclear how these relate to other clubroot populations estimated using the qPCR technique. Recently, *P. brassicae* occurred on the European winter *B. napus* cultivar “Mendel” and has given rise to worldwide outbreaks in *B. napus* crops. The screening and further development of clubroot-resistant varieties however are well-advanced, particularly in Europe. These developments are based on the transfer of effective genes from *B. campestris* using non-specific resistance in *B. campestris* and backcrossing with *B. napus* containing race-specific genes.

The success of these breeding programmes depends on accurate estimations of clubroot populations. Currently, this is based on plant differential sets [32]. Virulent pathotypes of *P. brassicae* capable of overcoming resistance have emerged in Alberta. However, relatively little attention has been given to the analysis of clubroot populations in soil used to characterise clubroot variation in differential set responses [33]. 

Using uniform characterised standards would be a useful attribute to similar breeding programmes.

Other members of the *Plasmodiophoraceae* family include other obligate root parasites. For example, the genus *Polymyxa* includes two species, *P. betae Keskin* and *P. graminis*, which are morphologically indistinguishable but separated by different host ranges. They can be distinguished by using a ribosomal DNA analysis. *P. betae* is also responsible for the transmission of viruses, such as beet necrotic yellow vein virus (BNYVV), the cause of rhizomania. The life cycle of *P. betae* is similar to that of *P. brassicae*, and there are some host-specific variations among *P. betae* isolates. Infection by virus-free *Polymyxa* spp. remains asymptomatic. Lateral root proliferation is observed following the transmission of BNYVV. Primers corresponding to sequences within the internal transcribed spacer region 2 (ITS2) of ribosomal DNA were used successfully to detect *P. betae* DNA using real-time PCR in soil samples. These studies also used plant-based cystosori dilution samples to extract DNA as standards used in the study. The use of plant-based standards for the detection of obligate pathogens is widespread and a major weakness in adapting standardised approaches for the detection of obligate pathogens.

The work described in this study details an alternative method of producing clubroot standards using recombinant technology. Clubroot DNA was recombinantly produced in a new expression system, which can be used as an alternative to clubroot-generated standards from gall tissues. The recombinant ITS region standard produced a single pure amplicon when amplified in a PCR reaction, reflecting the specificity of the primers for the ITS region. Amplicon purity helps to ensure that downstream applications can also be detected with a greater accuracy.

The external validation of the recombinantly produced standard at different concentrations is further evidence of the detection capability of the recombinant DNA. Most interestingly, this work also demonstrated a higher sensitivity at lower concentrations of the recombinantly produced standard.

Producing recombinant standards for clubroot can help to improve the sensitivity of the detection assays currently used. Low spore numbers may not pose an issue in the short term, but viable spores can quickly increase in concentration and populate small localised areas of land. The sensitivity of the recombinant clubroot standards at lower concentrations in this work indicates that these may be superior to DNA extracted from gall tissue for use as standards. The use of recombinantly produced standards presents a useful opportunity for clubroot researchers to move towards using a set of conserved DNA sequence standards for clubroot biotype detection.

The use of recombinantly generated standards also gives the benefit of being able to use specific DNA sequences, and these can then be used globally, allowing disease populations to be tracked more precisely. 

## 4. Methods

### 4.1. Method of Clubroot DNA Standard Production

This method followed the steps and modifications used in [34]. Galls were washed and cut into approximately 1 cm^3^ pieces. Distilled water was added to the gall pieces at a ratio of 5:1 water (mL) to gall tissue (g) and macerated with a hand blender. The suspension was filtered using a muslin cloth, which was folded over 4 times. The spore suspension was then centrifuged at 2500 rpm for 10 min, and the supernatant was discarded. The top layer of the pellet was removed by washing the pellet surface with ice-cold sterile distilled water containing 0.05% sodium azide.

Finally, the spores were resuspended to a concentration of 10^9^ using a haemocytometer and immediately mixed with air-dried soil. Clubroot DNA was subsequently extracted using a soil extraction kit, and it is described in the next section.

### 4.2. DNA Extraction

An M0Bio UltraClean Soil DNA isolation kit was used with a 0.3 g sample of each control and artificially infested soils, and it was processed according to the manufacturer’s instructions (MoBio Laboratories, PO Box 606 Solana Beach, CA, USA). To maximise the DNA isolation yields, an alternative protocol was followed, and a Fast Prep device (Qbiogene, Carlsbad, CA, USA) was used according to the manufacturer’s guidelines. For each soil, a 50 µL DNA volume was purified using a method supplied by Dr R. Faggian [35]. For each extracted DNA soil extract, 10-fold dilutions to 1 in 100 were made in a TE buffer (10 mM Tri-Cl, 1 Mm EDTA, pH 8.0). 

### 4.3. Preparation of DNA Sample Extracts for PCR

A 5 µL volume of isolated DNA from a soil sample extract was mixed with 15 µL of PCR Master Mix (11.35 µL H_2_O, 2.5 µ PCR buffer 10x, 0.75 µL 50 mg (1.5 mM) Mg, 0.2 µL DNTP, 0.2 µL DNase) and 30 ng of *P. brassicae* specific primers [33]. A negative control of molecular-grade water + PCR Master mix and primers was included. Employing a hybrid PCR thermal cycler machine, sample amplicon products (amplified specific sequence of sample DNA) were produced and, if present, visualised using agarose gel separation. Then, 2.5% Metaphor gels were used, and, for each PCR sample product, 8 µL was mixed and loaded with 3 µL of Xylene cyanol buffer. 

A molecular weight marker was applied to the gel at a 1µL sample volume. In a continuous ethidium bromide buffer, the electrophoretic separation of sample components at 150 v for 25 min was achieved. The fluorescent yield of the ethidium bromide–DNA amplicon complex was determined using a transilluminator, emitting light at 302 nm, and a photographic image was recorded.

### 4.4. Recombinant DNA Production

The sequence of the recombinant ITS DNA region fragment is shown below:

GCGGAAGGATCATTAACACAGTGGGCGGCCCTAGCGCTGCATCCCATATCCAACCCCATGTGAACCGGTGACGTGCGGCGACTCCAGCTGCGTGTTTCATTTTCGAACCATCCTAGCCGAAACACAACTAAAGTTCCATACATACATACATGTTACAACTCTTAGC (*Sourced from GenBank: AF231027.1*)

The above ITS DNA sequence was inserted into a pUC57 vector using the restriction enzyme Eco*RV*. Eco*RV* was used to digest the pUC57 vector and insert the ITS clubroot DNA region. The clubroot DNA fragments and linearised pUC57 vector were then ligated. The clubroot vector was then electroporated into *E. coli* TOP10 cells and left to grow at 37 °C for 12 h in LB. Selected plasmids were tested for the presence of the transformed vector. The extracted DNA was amplified and lyophilised. The production of the recombinant vector was carried out using Genscript, the Netherlands.

### 4.5. Recombinant ITS DNA PCR

The following reagents were added to a reaction tube: a high-fidelity buffer (1X final concentration), dNTPs (200 µM final concentration), ITS3 and ITS4 primers (0.5 µM final concentration), clubroot DNA at a final concentration of 100 ng, Phusion DNA polymerase 0.02 U/µL and H_2_O up to a final volume of 50 µL. The cycling conditions were carried out at the following temperatures and cycles: 95 °C for 5 min, 95 °C for 30 s and repeated for 45 cycles, 95 °C for 60 s, 72 °C for 60 s and, finally, 72 °C for 10 min. PCR samples were run on 1% agarose gel, with electrophoresis carried out at 90 V for 60 min. 

### 4.6. qPCR Protocol

The following reagents were added to a 96-well plate set up on ice: SYBR Green Universal supermix (Biorad, USA) at a volume of 10µL, representing a final concentration of 200 nM; 1µL of each forward and reverse ITS primers (200 nM final concentration) (ITS3-CGCTGCATCCCATATCCAA; ITS4 TCGGCTAGGATGGTTCGAAA); 2 µL DNA template; and 3 µL molecular-grade water. The cycling conditions were as follows: 95 °C for 5 min, followed by 45 cycles of 95 °C for 30 s, 95 °C for 60 s, 72 °C for 60 s and, finally, 72 °C for 10 min.

## Figures and Tables

**Figure 1 plants-12-01690-f001:**
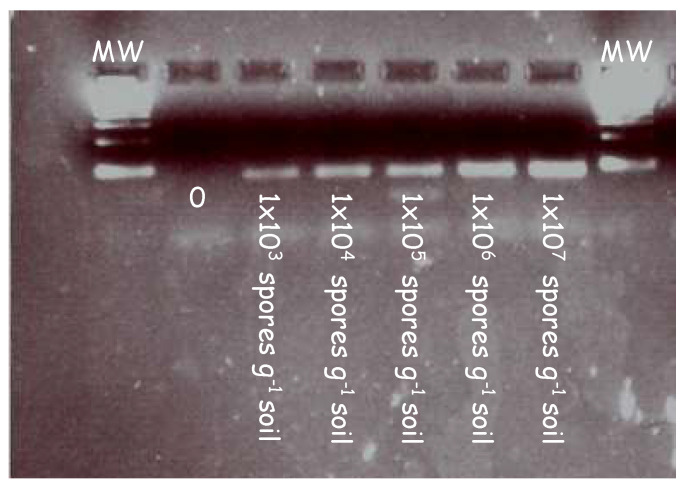
PCR detection of clubroot DNA extracted from samples artificially infested with increasing concentrations of clubroot in soil. Clubroot soil samples were amplified in a series of concentrations using primers targeting the internal transcribed region (ITS). A clear band was detected at each concentration ranging from 1 × 10^3^ to 1 × 10^7^. Samples were diluted 1/10 before being loaded into the gel and electrophoresed.

**Figure 2 plants-12-01690-f002:**
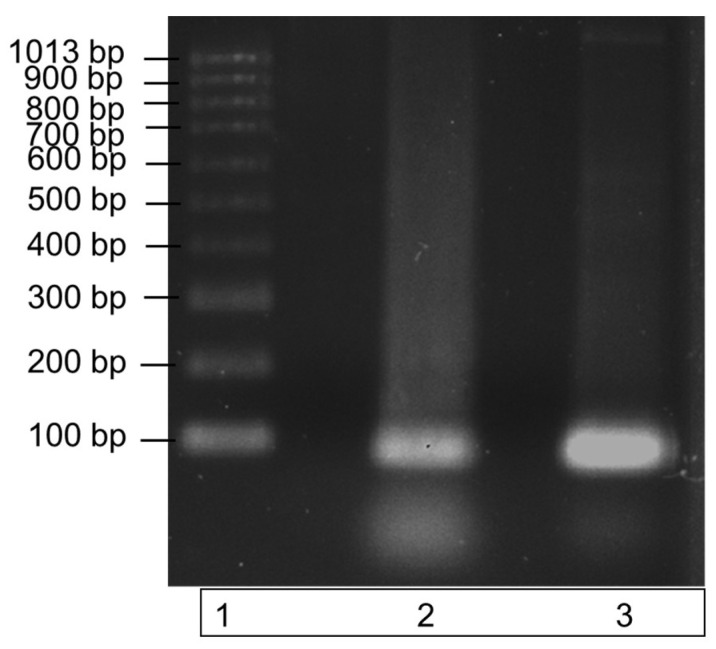
Amplification of recombinant ITS region DNA using PCR. The specificity of the ITS region primers to the recombinantly produced ITS DNA is illustrated by a single band at a size of 100 bp in lane 3. DNA bands were run on a 1% agarose gel for 70 min at 90 V. Lane 1: molecular weight ladder; lane 2: recombinant ITS region DNA (control); lane 3: recombinant ITS PCR amplicon.

**Figure 3 plants-12-01690-f003:**
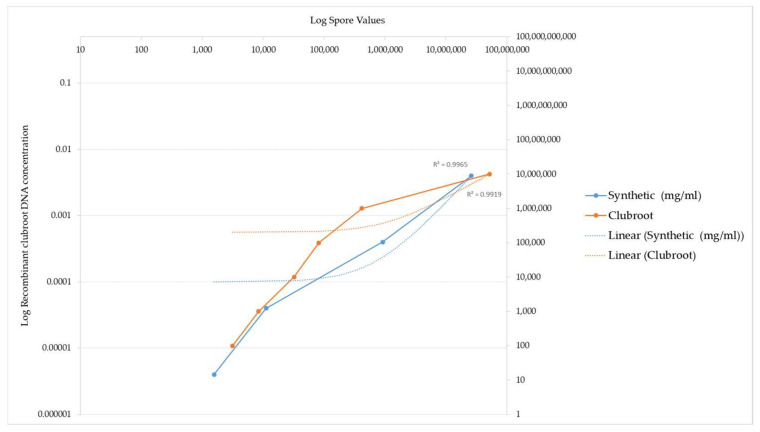
PCR amplification of recombinant clubroot DNA versus clubroot DNA from natural sources (galls). This graph demonstrates the sensitivity of recombinant clubroot DNA at increasing concentrations of DNA. Spore number represents the final amount of DNA amplified after the PCR reaction. The graph illustrates that the spore number increases as the concentration of recombinant DNA increases, indicating that the recombinant clubroot DNA can be amplified at a range of concentrations and is sensitive to different concentrations. For reference, a commercial clubroot standard is also included.

**Table 1 plants-12-01690-t001:** Comparison of the detection of recombinant ITS region DNA with DNA extracted from galls infected with clubroot. As the concentration of recombinant DNA increases, the spore number also increases. This relationship is also reflected by the infected root samples. Spore number increase correlates to higher spore concentrations. Final results are represented by mean spore numbers derived using a qPCR assay.

Recombinant ITS Region DNA (mg/mL)	Spore Number (Natural Clubroot Standard)	Root (mg/mL)	Spore Number (Clubroot Gall-Derived Standard)
0.000004	1569	CR 10^2^	3151
0.00004	11,171	CR 10^3^	8422
0.0004	915,303	CR 10^4^	32,211
0.004	26,174,274	CR 10^5^	81,783
0.04	4,881,103,992	CR 10^6^	417,797
0.4	18,924,239,046	CR 10^7^	52,022,765

## Data Availability

Information on the sequence information for the clubroot gene was sourced from the GenBank database on the NCBI website (www.ncbi.nlm.nih.gov). The information was accessed on 7 August 2020.

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
