# Peer review of "Production of Clubroot Standards Using a Recombinant Surrogate to Overcome Natural Genetic Variability"

_plants, 2023, doi:10.3390/plants12081690_

Round 1

Reviewer 1 Report

Major concerns:

At several points you make reference to “universal”.  There is pretty minimal detail given (at Line 227), but it would seem that you have only used 1 isolate of P. brassicae.  Considering that this is an important pathogen on 4 continents (Line 55-56) , you need to do a lot more validation before you can claim “universal”.  

You include Wallenhammar (2011) in your references, but I don’t see how your present work goes beyond that paper.  You state that the Wallenhammar work “…hasn’t been validated against clubroot DNA extracted from natural sources” but it looks like that paper included >80 soil samples from 7 fields at 3 different farms.  Your work seems to include 

Most critically, I don’t see the goal of this paper.  The stated aims, that is, the problem they seem to be trying to solve, seems to be already addressed in the literature.  The introduction is needlessly long and whatever point the authors need to get to is lost.  

Minor points:

The abstract contains unnecessary material – Line 9-10 are better suited for the intro.  Lines  13-20 can probably be condensed to 1-2 sentences.  

Line 46:  What’s the linkage between temperature and spread of clubroot disease?

Line 47:  Why capitalize “Lime”?

Line 50:  “newer strains”?  Not clear if you are talking about the host or the pathogen here.

Line 62:  the “…most…compared to…” sentence construction is poor.  Probably should read “…are more successful than other detection methods”

Line 88:  I don’t understand this point.  

Line 94:  Unclear why the DNA is unstable.  What do you mean here?  

Line 111:  The reference you have addresses the variability of the ITS region in Fusarium.  You are not working with Fusarium, or even an organism in the same Kingdom.  

Figure 1:  Add sizes for the MW marker lane.

Line 227:  We need a LOT more information here.  “Mature galls” from what?  What was the host plant?  Was it grown in the greenhouse or in the field?  Artificially inoculated?  What was the source of the inoculum?  

Line 234:  What was the nature of the soil?  Was there an extended incubation or did you just mix spores into the soil and then extract?  

Line 283:  This protocol is unclear – does 10uL indicate the volume of the supermix or of the total reaction?  What does the 200nM  in line 283 refer to?

Author Response

At several points you make reference to “universal”.  There is pretty minimal detail given (at Line 227), but it would seem that you have only used 1 isolate of P. brassicae.  Considering that this is an important pathogen on 4 continents (Line 55-56) , you need to do a lot more validation before you can claim “universal”.

Universality refers to a common approach being adopted by all practitioners for the method of synthetic clubroot DNA production. The basic point of any standard is the breadth of its commonality against the context that currently exists, clubroot standards are made from galls sourced from different locations which therein must and does creates genetic variability between consequently variable clubroot standards. The differential sets used to differentiate isolates are different in many countries and therefore not comparable.  This is another reason why standardisation is necessary since quantification will be sequence related.

The ITS region is a conserved region across different Clubroot genomes. Synthetic production of this region and its use in clubroot assays could help the field move towards a universal approach.

We could refer to UK only if that would help. 

You include Wallenhammar (2011) in your references, but I don’t see how your present work goes beyond that paper.  You state that the Wallenhammar work “…hasn’t been validated against clubroot DNA extracted from natural sources” but it looks like that paper included >80 soil samples from 7 fields at 3 different farms.

In the Wallenhammar paper the lowest concentrations of their ‘standards curve’ had been extrapolated from the data set they had obtained.

Sensitivity of the lowest concentration of clubroot standards presented in this work is higher. This allows earlier detection of Clubroot and/or less likely for its presence to be missed.

The approach used by Wallenhammer was not fully tested and the sequence used was never fully investigated in the isolates as her work predated second generation sequencing and was therefore too difficult and expensive to complete. Therefore we believe that there is little comparability

Most critically, I don’t see the goal of this paper.  The stated aims, that is, the problem they seem to be trying to solve, seems to be already addressed in the literature.  The introduction is needlessly long and whatever point the authors need to get to is lost.

Standardisation of clubroot standards will help epidemiological work go forward on clubroot.  We can suggest altering the introduction to stress this point and also the title.

Reviewer 2 Report

This protocol is a step towards overcoming an old problem, namely that clubroot standards cannot be prepared with high reproducibility for quantification. Therefore, this work is of interest. Some minor changes are recommended. In the abstract line 9 add "for entry". Lines 11, 12, add"and other economically important brassica crops". Introduction line 69: should start with capital letter. At the end of introduction the authors discuss what are problems with DNA from viable vs non-viable spores, but their method does not provide any solution for that. This should be described more cautiously. Same for the discussion. The model should be better explained. Results line 111 the authors mention 'fungi', even though it is assumed that they do not mean Plasmodiophora, it should be clearly stated that this is a protist and not fungus, even though ITS from fungi are mentioned.

Author Response

This protocol is a step towards overcoming an old problem, namely that clubroot standards cannot be prepared with high reproducibility for quantification. Therefore, this work is of interest. Some minor changes are recommended. In the abstract line 9 add "for entry". Lines 11, 12, add"and other economically important brassica crops". Introduction line 69: should start with capital letter. At the end of introduction the authors discuss what are problems with DNA from viable vs non-viable spores, but their method does not provide any solution for that. This should be described more cautiously. Same for the discussion. The model should be better explained. Results line 111 the authors mention 'fungi', even though it is assumed that they do not mean Plasmodiophora, it should be clearly stated that this is a protist and not fungus, even though ITS from fungi are mentioned.

Line 9: Added ‘for entry’

Line 11-12: Other economically brassica crops added

Line 69: Clubroot capitalised

Line 99: Additional sentence in response to viable/non-viable query: Information on total resting spore numbers in soil (both viable and non viable) would be useful.

Line 111: Fungi changed to protist

The approach seem unique for quantification of clubroot pathogen resting spores, which could potentially be of interest to a broad range of audience. Likely its biggest application is on quantification of clubroot soil inoculum pressure, but it seems that the study was based only on pure resting spores and it would be an important validation if spiked soil samples can be tested.

The introduction can be better focused; highlight current methodologies used for inoculum quantification and deficiencies, and other aspects of disease can be brief or omitted.

Lines 49-60 related to disease severity and these have been deleted.

"Recombinant clubroot concentration standards" or similar terms appear throughout the manuscript. Clubroot concentration is unclear, and it should be referred to as the concentration of pathogen spores, inoculum or DNA.

Recombinant concentration changed to DNA concentration throughout.

Figure 3: Some stats value would be helpful; regression analysis? correlation coefficients?

R values have now been included.

Line 226: Method of clubroot protein standard production -Confusing. Are you producing recombinant DNA or protein?

This has been modified to ‘Method of clubroot DNA standard’.

Round 2

Reviewer 1 Report

In the author's response it was proposed that the term 'universal' be omitted, but the change was not made.  You have not validated this beyond 1 strain so 'universal' is not acceptable.  Have you done a Genebank search to show the level of conservation /specificity of this sequence?  Until that work is done AND shown, you can't say universal.

Author Response

Thank you for taking time the time to review this paper.

The term universal has now been omitted from the discussion and is a fair evaluation.

Round 3

Reviewer 1 Report

Thanks for making the change.

Author Response

The highlighted regions of the text represent where the additional references have been added. The end of the introduction and all of the discussion have been modified.

The primary concern was use of 'universal standard' which has now been omitted. There are also further justifications for the purpose of the paper.
